# Kisspeptin improves local ovarian insulin resistance in PCOS by modulating the PI3K/AKT/GLUT4 signaling pathway

Na Sun, Pingping Sun, Lilan Sun, Ping Sun, Dandan Qi, Huagang Ma 🆔*

Reproductive Medicine Center, Weifang People's Hospital, Weifang, Shandong Province, China

* mahuagang8192693@163.com

## Abstract

### Background

Insulin resistance (IR) is commonly observed in patients with polycystic ovary syndrome (PCOS), affecting 44% to 70% of these individuals. Kisspeptin is a key regulatory factor in energy balance and reproduction, and it may alleviate PCOS-related symptoms by improving insulin resistance.

### Methods

In this study, a PCOS-IR mouse model was established using dehydroepiandrosterone (DHEA) and a high-fat diet. The expression of kisspeptin, PI3K, phosphorylated PI3K (p-PI3K), AKT, phosphorylated AKT (p-AKT), and glucose transporter 4 (GLUT4) was measured by immunofluorescence staining, quantitative PCR, and Western blotting. Flow cytometry was used to evaluate mitochondrial membrane potential (MMP) and reactive oxygen species (ROS) levels.

### Results

In granulosa cells from PCOS-IR mice, kisspeptin upregulated GLUT4 expression by activating the PI3K/AKT signaling pathway. In vitro experiments showed that kisspeptin significantly reduced ROS levels, enhanced MMP, and improved mitochondrial function.

### Conclusion

Kisspeptin improves insulin resistance through the PI3K/AKT/GLUT4 signaling pathway and exerts its effects in vitro in granulosa cells. by protecting mitochondrial function. This study provides potential biomarkers and therapeutic targets for the treatment of PCOS-IR.

**Data availability statement:** All relevant data are within the manuscript and its Supporting Information files.

**Funding:** This work was supported by the Weifang Health Commission (Grand number: WFWSJK-2023-005). The funders had no role in study design, data collection and analysis, decision to publish, or preparation of the manuscript.

**Competing interests:** The authors have declared that no competing interests exist.

## Introduction

Polycystic ovary syndrome (PCOS) is a prevalent endocrine disease of the reproductive system, often accompanied by metabolic disorders [1]. A critical factor in this condition is the presence of insulin resistance (IR), which disrupts normal glucose transport functions. Reports indicated that the prevalence rates of insulin resistance in PCOS patients range from 44% to 70% [2]. Insulin facilitates glucose metabolism by activating the classic signaling pathway IRS-1/PI3K/AKT, where the expression and translocation of glucose transporter 4 (GLUT4) to the membrane enhance cellular glucose uptake, thereby maintaining blood glucose homeostasis [3]. In patients with PCOS, the PI3K/AKT pathway was inhibited in the ovaries, resulting in decreased expression of the downstream protein GLUT4, which was closely associated with the development of insulin resistance [4,5]. However, some literature suggests insulin sensitivity in ovary under hyperandrogenic conditions with activation of insulin pathways [6].

Interestingly, despite systemic insulin resistance, certain ovarian compartments—particularly granulosa and theca cells—may exhibit preserved or even heightened insulin sensitivity. This paradox has been attributed to tissue-specific differences in insulin signaling pathway regulation [7].

In peripheral tissues such as skeletal muscle and adipose tissue, insulin resistance primarily involves impaired IRS-1 phosphorylation and attenuated PI3K/AKT signaling, leading to reduced GLUT4 translocation and glucose uptake. In contrast, ovarian cells can retain partial responsiveness to insulin, and the mitogen-activated protein kinase (MAPK) pathway may remain active or become selectively upregulated [8]. This differential regulation allows insulin to continue stimulating androgen synthesis via the MAPK/ERK cascade in theca cells, even when metabolic signaling through PI3K/AKT is compromised [9]. Such divergence in pathway activation underscores the complexity of insulin action in PCOS and highlights the ovaries as both a target and a contributor to the syndrome's endocrine–metabolic interplay.

Kisspeptin is a neuropeptide encoded by the Kiss1 gene and secreted by the hypothalamus [10]. The interaction between kisspeptin and its receptor (KISS1R) plays a critical role in activating the hypothalamic-pituitary-gonadal (HPG) axis, which is essential for the onset of puberty and the maintenance of reproductive functions [11]. Kisspeptin serves as a link between energy homeostasis and reproduction, as kisspeptin neurons were in close proximity to leptin and insulin receptors, allowing them to receive energy signals and influence reproductive processes [12]. Furthermore, kisspeptin and KISS1R were expressed in various peripheral tissues, including pituitary, pancreas, adipose tissue, and reproductive tissues [10]. Women with PCOS exhibited abnormal kisspeptin levels in both serum and follicular fluid, which may impact follicular development and ovulation [13]. While in vitro and in vivo studies have demonstrated that kisspeptin stimulates insulin secretion, the existing literature provides limited and conflicting evidence, preventing a definitive conclusion regarding the relationship between kisspeptin and insulin resistance in the ovaries of individuals with PCOS [14].

Mitochondrial dysfunction and oxidative stress are considered important mechanisms underlying insulin resistance [15]. Excessive production of reactive oxygen species (ROS) and a decrease in mitochondrial membrane potential (MMP) can disrupt insulin signaling, impair cellular energy metabolism, and promote granulosa cell dysfunction [16]. Widespread structural and functional abnormalities of mitochondria in the ovarian tissues of patients with PCOS as well as in animal models, suggesting that mitochondrial damage may represent a critical link between metabolic disturbances and reproductive dysfunction.

However, it remains unclear whether kisspeptin can alleviate PCOS-associated insulin resistance by regulating the PI3K/AKT/GLUT4 signaling pathway and improving mitochondrial function. Therefore, in the present study, a PCOS-IR mouse model was established using dehydroepiandrosterone (DHEA) combined with a high-fat diet to systematically investigate the regulatory effects of kisspeptin on insulin signaling and mitochondrial function in granulosa cells, with the aim of identifying potential biomarkers and therapeutic targets for the prevention and treatment of PCOS-IR.

## Methods

### PCOS-IR model

Animal experiments were conducted in accordance with the guidelines of the Experimental Animal Ethics Committee. Female C57/BL6J mice(21-day-old)(Vital River, China) were raised under standard laboratory conditions. They were randomly divided into two groups. The establishment of the PCOS-IR model was based on the study by Li et al [17].

Animals in the DHEA groups (12 mice) were injected daily with dehydroepiandrosterone (GC11070, GlpBio,China) (DHEA, 6 mg/100 g body weight, dissolved in 0.2 mL of sesame oil) and given a high-fat diet to induce PCOS-IR. Mice in the control groups (6 mice) were injected daily with equal amounts of sesame oil and given a normal diet. After 20 days of continuous treatment, the mice were fasted for 12 h before the glucose tolerance test (GTT). Blood glucose detection equipment (Roche, China) was used to measure glucose levels in tail vein blood. After fasting, fasting glucose levels were first measured. Mice were then given an intragastric infusion injection of 2 g/kg body weight glucose, with tail samples 0,15,30,60 and 120 min afterward. At the end of the experiment, euthanasia was performed on the mice by gradually filling the chamber with $CO_2$.

### Haematoxylin and eosin (H&E) staining of the ovaries

After fixation with 4% paraformaldehyde (G1101, Servicebio, Wuhan, China), the ovaries were trimmed, dehydrated, embedded, sectioned, stained with haematoxylin and eosin (C0105S, Beyotime, China), sealed, and then observed under a microscope (Pannoramic SCAN II, 3DHISTECH, China), and images were taken.

### Enzyme-Linked Immunosorbent assay

After the completion of model establishment (20 days of treatment), testosterone levels were measured. The levels of testosterone (T)(PT872, Beyotime, Shanghai, China) and fasting insulin (INS)(PI602, Beyotime, Shanghai, China) in mouse serum were determined by enzyme-linked immunosorbent assay kits for mice.

### Ovulation induction and granulosa cells collection

Mice were injected with 10 units of pregnant horse serum gonadotropin (PMSG)(Solarbio, China). After 72h, the cervical vertebra was dislocated and killed. Bilateral ovaries were quickly removed and immediately placed in PBS in petri dish under aseptic conditions. The process is as follows: puncture large follicles with a 1 mL syringe needle under anatomic microscope and release granulosa cells into DMEM-F12 medium(DMEM F/12 + 10%FBS + 1%PSS + 1% ITS + 2% L-Glutamine)(CM10090, MACGENE, Beijing, China), disperse cells into single-suspended cells by pipetting in centrifuge tubes, filter the cells with a 70um filter cloth, centrifuge at 1000 rpm for 5 min, discard the supernatant, resuspend cells in culture medium, counted and plated in 24-well plate at a density of $3 \times 10^4$ cells/well.

## Immunofluorescence staining

All antibody details are listed in Table 1. The procedure was carried out at room temperature unless otherwise noted. Granulosa cells were plated into 24-well plates and allowed to grow into a monolayer. We follow these steps: fix the cells in 4% paraformaldehyde(G1101, Servicebio, Wuhan, China) for 20 min and wash three times with PBS for 5 minutes each time. Add 300μl of 0.5% Triton x-100(T8200, Solarbio, Beijing, China) to each well and incubated for 20 min, then wash three times with PBS. Drop normal goat serum(SL038, Solarbio, Beijing, China) on the slide, and seal it for 30 min. Incubate sections with primary antibody overnight at 4°C and then wash in PBS for 5 minutes three times. Incubate with secondary antibody for one hour and wash in PBS for 5 minutes three times. Add blue fluorescent antibody DAPI(C0065, Solarbio, Beijing, China), incubate for 10 min in dark, and wash three times with PBS. Drop anti-fluorescence quencher onto glass slides, covered with a cover glass. A fluorescence microscope (BX50, Olympus, Tokyo, Japan) was used to observe and collect fluorescence images.

## Drug stimulation

The granulosa cells at the logarithmic phase were prepared into a single-cell suspension after digesting by trypsin(Solarbio, China) that was subsequently seeded into a six-well plate. Following the overnight culture, the medium was changed to a serum-low medium. Cells were then stimulated with 10ng/ml Kisspeptin(MCE, China) or 100nm KISS1 receptor antagonist (Kisspeptin 234 TFA) (MCE, China). After drug treatment for 48 hours, cells were collected.

## Western blot

Rinsed the treated cells with PBS and added appropriate amount of 1×SDS-PAGE protein loading buffer(Beyotime, China), then used a cell scraper to scrape the cells to boil on a metal bath at 100°C for 5–10 min. Polyacrylamide gel was prepared according to the protocol of the SDS-PAGE Gel Kit(P0012A, Beyotime, Shanghai, China). Samples were subjected to electrophoresis, membrane transfer, and blocking. Proteins (60 μg) were separated through 8–10% sodium dodecyl sulphate–polyacrylamide gel electrophoresis and transferred onto polyvinylidene fluoride membranes (Millipore, USA). After blocking with 5% non-fat milk, the blots were incubated with primary antibodies (Supplementary Table 2) against PI3K, phosphorylated-PI3K(p-PI3K), AKT, phosphorylated-AKT(p-AKT), GLUT4(1:1000; Abcam, USA). Subsequently, the protein bands were incubated with HRP-conjugated secondary anti-rabbit antibody or anti-mouse antibody (1:2000; Abcam, USA) (Table 2) and visualised after incubation with enhanced chemiluminescent reagent (Millipore, USA). This experiment was repeated six times.

## ROS generation

The cells were harvested by trypsin digestion and centrifugation. DCFH-DA (S0035S, Beyotime, Shanghai, China) was diluted 1:1000 with serum-free medium to a final concentration of 10 μmol/L. The cell culture medium was removed and an appropriate volume of diluted DCFH-DA was added and incubated in a 37 °C cell incubator for 20 min. The cells were washed three times with a serum-free cell culture medium to fully remove the DCFH-DA that had

**Table 1. Information on the primary and secondary antibodies involved in immunofluorescence staining.**

| Primary antibodies | | |
|---|---|---|
| GLUT4 | HUABIO | R1402-3 |
| KISS1 | Proteintech | 18375-1-AP |
| Secondary antibodies | | |
| FITC-labeled goat anti-rabbit IgG | Servicebio | GB22303 |
| Cy3-labeled goat anti-rabbit IgG | Servicebio | GB21303 |

**Table 2. Information on the primary and secondary antibodies involved in western blot.**

| Primary antibodies | | |
|---|---|---|
| actin | ZSGB | TA-09 |
| PI3K | Proteintech | 67071-1-Ig |
| AKT | Proteintech | 10176-2-AP |
| P-PI3K | HUABIO | HA721673 |
| P-AKT | Proteintech | 80455-1-RR |
| GLUT4 | HUABIO | R1402-3 |
| Secondary antibodies | | |
| anti-mouse antibody | ZSGB | ZB-2305 |
| anti-rabbit antibody | ZSGB | ZB-2301 |

not entered the cells. After resuspension, the cells were subject to flow cytometry analysis (CytoFLEX, BECKMAN, Brea, California, USA).

### Mitochondrial membrane potential detection

JC-1 working solution(C2005, Beyotime, Shanghai, China) consists of 50 µl JC-1 (200x), 8 ml ultrapure water, and 2 ml JC-1 staining buffer (5x). After washing with PBS, digestion and centrifugation, the cells were added 1 mL of JC-1 staining working solution, mixed well and incubated in the incubator at 37 °C for 20 minutes. Add JC-1 Staining Buffer (5x) to 4 mL of distilled water, take an appropriate amount of JC-1 Staining Buffer (1x), and place in an ice bath during incubation. After the incubation, discard the supernatant and wash twice with JC-1 staining buffer (1X). Resuspend cells and flow cytometry (CytoFLEX, BECKMAN, Brea, California, USA) test. This experiment was repeated six times.

### Statistical analysis

GraphPad Prism (version 9.5, GraphPad Software, San Diego, California, USA) was used to analyze data. Continuous variables were compared using the Student t test or analysis of variance. Results were considered statistically significant at $P < 0.05$.

## Results

### Abnormal glucose metabolism was present in ovarian granulosa cells of PCOS-IR mice

We established the normal-diet mouse models as a control group. After the induction of PCOS, H&E staining showed a decrease in the number of antral follicles, while the number of cystic follicles increased significantly (Fig 1A). DHEA mice were injected with DHEA and fed a high-fat diet to exaggerate PCOS phenotypes, which had significantly elevated serum testosterone levels($P < 0.01$, Fig 1A). Fasting glucose and OGTT levels were higher in PCOS mice than the control group (Fig 1C). The area under the curve showed a significant difference in blood glucose levels between the two groups (Fig 1D, $P < 0.01$). Fasting insulin and insulin resistance index(HOMA-IR) were also significantly increased in PCOS mice ($P < 0.01$, Fig 1D-E).

### Kisspeptin decreased in ovarian granulosa cells of PCOS-IR mice

Ovarian granulosa cells from control group and PCOS-IR Model group are shown in Fig 2A. GLUT4 expression and translocation in the model group to the membrane were reduced ($p < 0.001$) (Fig 2B). The results of immunofluorescence staining showed a significant decrease in the expression of kisspeptin in the Model group ($p < 0.001$) (Fig 2C).

 

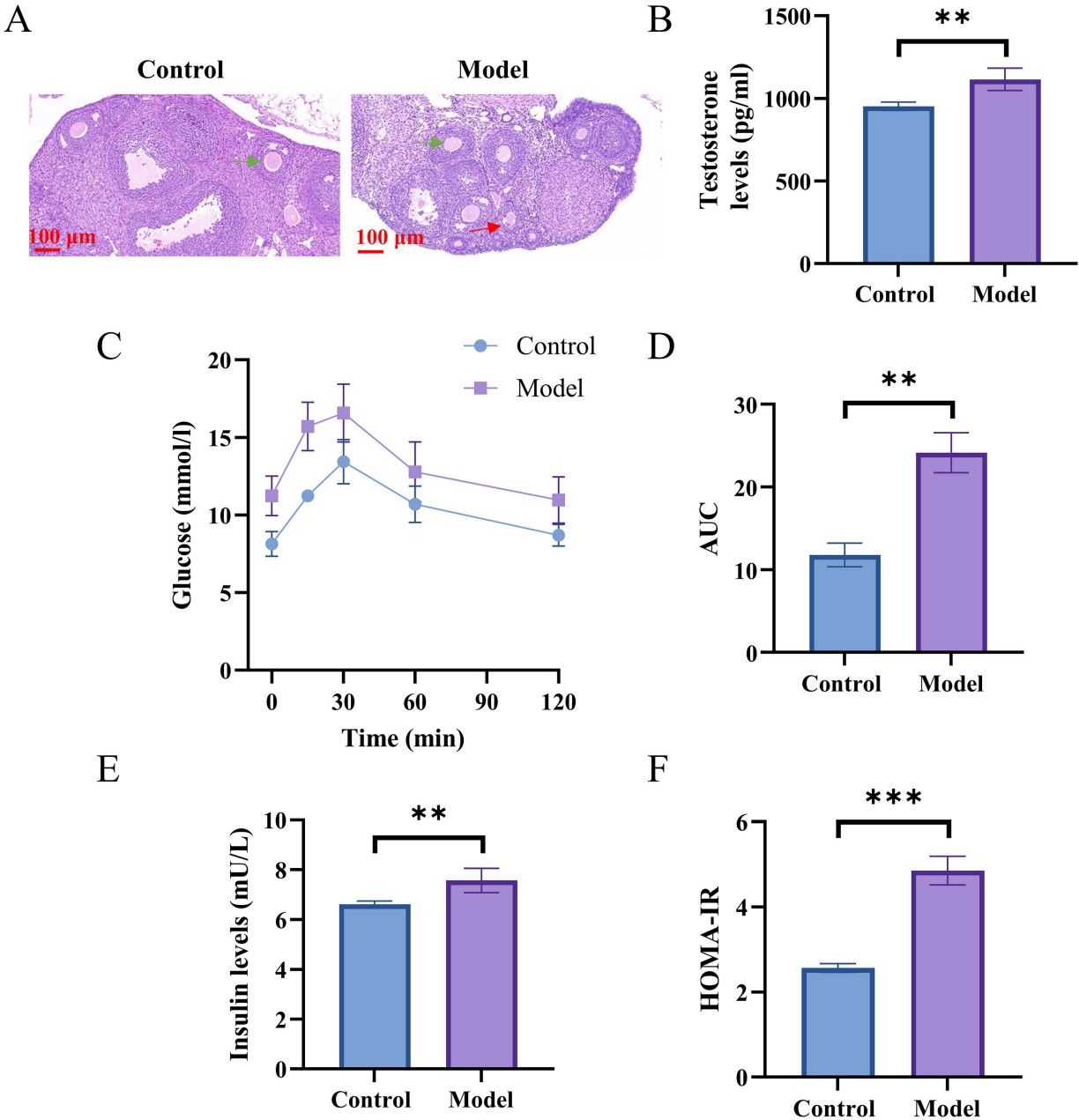

**Fig 1. Control group was injected with sesame oil and given a normal diet, Model group was injected with DHEA and given a high-fat diet.** After establishing the PCOS-IR model, the mice were sacrificed on day 20. **A.** H&E staining of ovaries sections. The green arrow indicates an antral follicle, and the red arrow indicates an ovarian cyst. **B.** ELISA to measure serum testosterone. **C.** The level of the fasting glucose. **D.** Area under the curve (AUC) for fasting plasma glucose. C-D. ELISA to detect insulin. **E.** ELISA to measure insulin. **F.** HOMA-IR = Fasting glucose (FPG,mmol/L) ×Fasting insulin (FINS,mU/L)/22.5. n = 6. **p < 0.01, ***p < 0.001.

## Kisspeptin activated the PI3K/AKT/GLUT4 signaling pathway

We stimulated granulosa cells from PCOS-IR mice with 10ng/ml kisspeptin or 100nm KISS1 receptor antagonist (Kisspeptin 234 TFA) for 48 hours by reference to previous literature. Compared with the PCOS-IR group, kisspeptin treatment

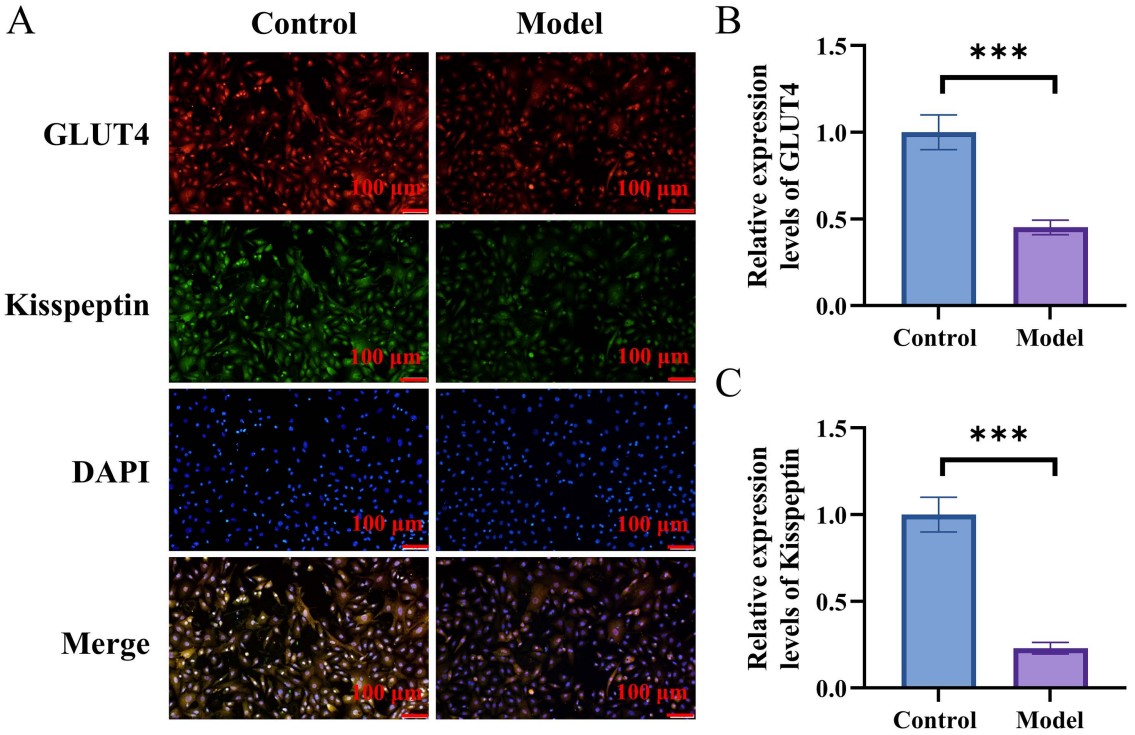

**Fig 2. Primary ovarian granulosa cells from control group and model group. A.** Immunofluorescence staining to test kisspeptin and GLUT4. **B.** The quantitative expression level of GLUT4. **C.** The quantitative expression level of Kisspeptin. n=6. ***p<0.001.

significantly increased the protein levels of GLUT4, p-PI3K, and p-AKT (p<0.05, p<0.01, p<0.001), while treatment with Kisspeptin 234 TFA significantly reduced the protein levels of GLUT4, p-PI3K, and p-AKT (p<0.01, p<0.001). It suggested that kisspeptin could activated the PI3K/AKT/GLUT4 signaling pathway (Fig 3).

### Kisspeptin significantly improved mitochondrial function by reducing ROS levels and increasing mitochondrial membrane potential of PCOS-IR granulosa cell

To detect the effects of kisspeptin on mitochondrial function in the PCOS-IR granulosa cells, we used flow cytometry to assess mitochondrial reactive oxygen species and mitochondrial membrane potential. The results of Fig 4A-B show that kisspeptin significantly inhibited ROS production, while Kisspeptin 234 TFA significantly increased ROS production (p<0.001). Fig 4C-D shows that kisspeptin significantly increased mitochondrial membrane potential (MMP), whereas Kisspeptin 234 TFA significantly decreased MMP(p<0.001).

### Discussion

In this study, we systematically investigated the role of kisspeptin in regulating local ovarian insulin resistance in a PCOS-IR mouse model and its underlying mechanisms. Our results demonstrated that abnormal glucose metabolism and insulin resistance were present in PCOS mice, as evidenced by elevated fasting glucose, impaired OGTT, increased fasting insulin levels, and a higher HOMA-IR index. Meanwhile, ovarian granulosa cells exhibited reduced GLUT4 expression and impaired membrane translocation, together with increased oxidative stress and mitochondrial dysfunction. These findings are consistent with previous studies reporting that PCOS is characterized by systemic and local ovarian insulin resistance, defective PI3K/AKT signaling, GLUT4 dysregulation, and mitochondrial impairment in granulosa cells. Thus,

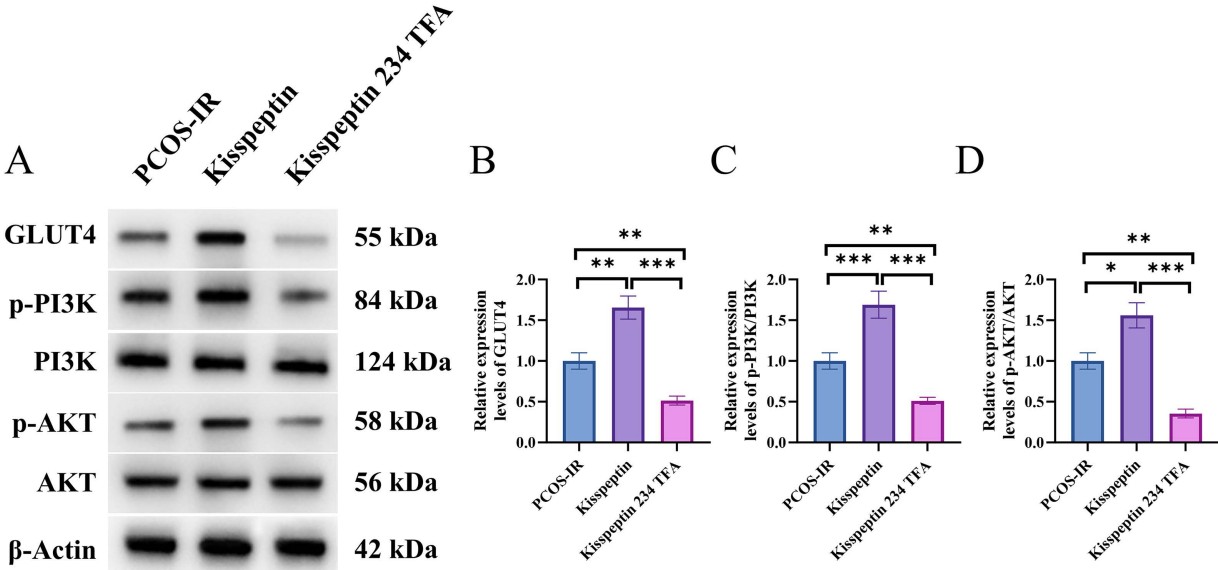

**Fig 3. PCOS-IR group:no drug stimulation to primary ovarian granulosa cells from PCOS-IR mice, Kisspeptin group: added kisspeptin, Kisspeptin 234 TFA group: added KISS1 receptor antagonist. A.** The protein expression levels of PI3K, AKT, p-PI3K, p-AKT, and GLUT4 were determined by Western blot. **B.** Quantitative analysis of the relative expression level of GLUT4 protein. **C.** Quantitative analysis of the relative expression level of p-PI3K/PI3K protein. **D.** Quantitative analysis of the relative expression level of p-AKT/AKT protein. $n = 6$. *$p < 0.05$, **$p < 0.01$, ***$p < 0.001$.

our data further confirm the reliability of the PCOS-IR model and support the critical involvement of metabolic disorders in PCOS pathogenesis [18–20].

Consistent with findings from recent studies, our study demonstrated that the expression of kisspeptin was significantly decreased in ovarian granulosa cells of PCOS-IR mice [21]. Previous studies have suggested that kisspeptin is not only a key regulator of the hypothalamic–pituitary–gonadal (HPG) axis but may also be involved in peripheral metabolic regulation and the maintenance of insulin sensitivity [22,23]. Our results further support the important role of kisspeptin in the cross-regulation between reproduction and metabolism.

However, unlike most previous studies that mainly focused on serum kisspeptin levels or its regulatory effects on the central reproductive axis, our study placed greater emphasis on the local ovarian microenvironment by directly detecting and validating changes in kisspeptin expression and its biological functions in granulosa cells [10,24]. Given that clinical studies have reported inconsistent conclusions regarding alterations in peripheral blood kisspeptin levels in patients with PCOS, our findings provide more direct and stable evidence at the tissue and cellular levels, highlighting the importance of the local regulatory role of kisspeptin.

At the mechanistic level, an important innovation of this study is that we, for the first time, directly demonstrated in PCOS-IR ovarian granulosa cells that kisspeptin improves insulin resistance by activating the PI3K/AKT/GLUT4 signaling pathway. Although previous studies have suggested that kisspeptin may be involved in the regulation of glucose metabolism, most of them remained at the level of correlative observations and lacked clear evidence of specific signaling pathways [25,26]. Our results showed that kisspeptin significantly upregulated the protein levels of p-PI3K, p-AKT, and GLUT4, whereas the KISS1R antagonist exerted opposite effects, thereby mechanistically establishing a direct regulatory role of kisspeptin in ovarian insulin signaling.

In addition, this study further links kisspeptin to the protection of mitochondrial function. Previous studies have confirmed the presence of mitochondrial dysfunction in granulosa cells of PCOS, characterized by increased reactive oxygen species (ROS) production and decreased mitochondrial membrane potential, which is consistent with our findings [27,28]. More

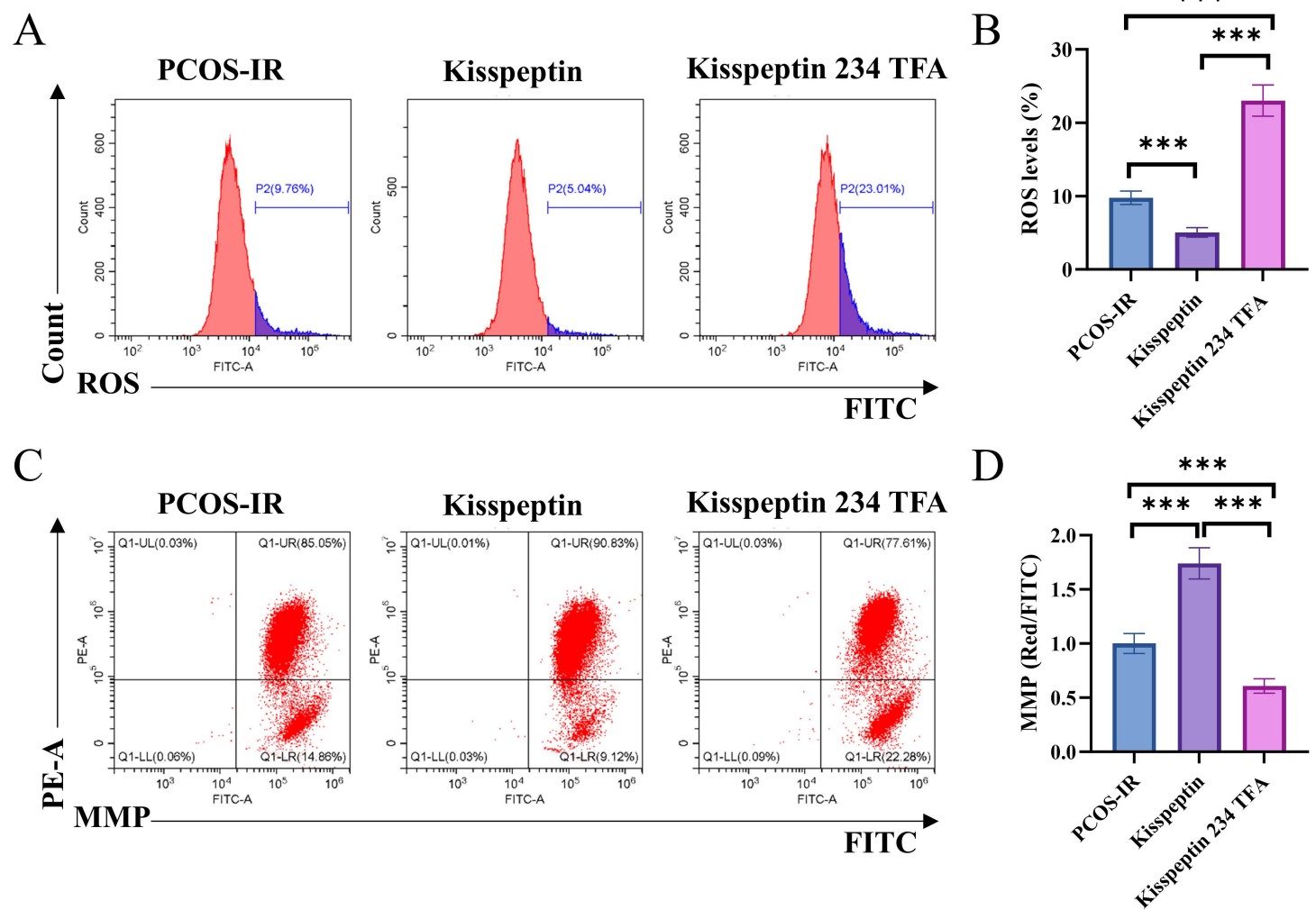

**Fig 4. ROS and mitochondrial membrane potential (MMP) were assessed using flow cytometry.** PCOS-IR group:no drug stimulation to primary ovarian granulosa cells from PCOS-IR mice, Kisspeptin group: added kisspeptin, Kisspeptin 234 TFA group: added KISS1 receptor antagonist. **A-B**. ROS levels in granulosa cells of PCOS-IR were measured by flow cytometry. **C-D**. MMP in granulosa cells of PCOS-IR were measured by flow cytometry. n=6. *p<0.05, **p<0.01, ***p<0.001.

importantly, we found that kisspeptin significantly reduced ROS generation and increased mitochondrial membrane potential, whereas the KISS1R antagonist aggravated mitochondrial damage. These results suggest that kisspeptin provides metabolic and energetic protection to ovarian cells by improving mitochondrial function. This dual mechanism of "insulin signaling regulation plus mitochondrial function improvement" has rarely been systematically reported in previous studies.

In terms of study design, this research combined a PCOS-IR animal model with in vitro experiments using ovarian granulosa cells, and adopted a bidirectional intervention strategy with both a kisspeptin agonist and a KISS1R antagonist. This approach strengthened the causal evidence beyond simple correlations and enhanced the reliability of the conclusions.

In summary, the similarities between our study and related research are mainly reflected in the following aspects: Both confirm that PCOS is closely associated with ovarian insulin resistance, inhibition of the PI3K/AKT pathway, dysfunction of

GLUT4, and mitochondrial damage; Both support the critical role of kisspeptin as a key link between reproductive regulation and metabolic regulation.

The main differences and innovations of this study are as follows:

It emphasizes the regulatory role of kisspeptin in the local ovarian microenvironment rather than only in peripheral circulation; It is the first to directly demonstrate in ovarian granulosa cells that kisspeptin improves local insulin resistance through the PI3K/AKT/GLUT4 signaling pathway;

It reveals the protective effect of kisspeptin on mitochondrial function, forming a "dual-pathway" synergistic regulatory mechanism; By combining an animal model with bidirectional receptor agonist/antagonist interventions, it significantly strengthens the mechanistic credibility of the conclusions.

Therefore, this study not only validates and extends existing theories regarding insulin resistance and mitochondrial dysfunction in PCOS, but also systematically elucidates the crucial role of the kisspeptin/KISS1R signaling pathway from the perspective of local ovarian metabolic regulation, providing stronger experimental evidence for kisspeptin as a potential therapeutic target for PCOS.

Despite the significant findings of this study, several limitations should be acknowledged. First, although the DHEA combined with high-fat diet mouse model effectively mimics the characteristics of PCOS with insulin resistance, physiological and metabolic differences between mice and humans may limit the direct translation of the results to clinical settings. Second, the current study mainly focused on in vitro experiments using ovarian granulosa cells, lacking validation in a more complex in vivo environment that reflects systemic metabolic regulation. Additionally, only a single concentration and a fixed stimulation time (10 ng/mL kisspeptin or 100 nM KISS1 receptor antagonist for 48 hours) were used. The absence of a dose-response or time-course analysis may overlook the dynamic effects of kisspeptin under different conditions. Furthermore, while the study confirmed that kisspeptin improves insulin resistance through activation of the PI3K/AKT/GLUT4 signaling pathway, the detailed downstream molecular mechanisms and the involvement of other signaling pathways remain to be elucidated. The relatively limited sample size in certain experiments, such as immunofluorescence, flow cytometry, and Western blotting, may also affect the robustness and generalizability of the findings. Finally, the study did not include validation using clinical samples from PCOS patients, and thus, the potential clinical application of kisspeptin in PCOS-IR requires further investigation.

## Conclusions

This study demonstrated that kisspeptin expression is reduced in ovarian granulosa cells of PCOS-IR mice. Kisspeptin improves insulin resistance by activating the PI3K/AKT/GLUT4 signaling pathway, enhancing GLUT4 expression and translocation. It also reduces ROS levels, increases mitochondrial membrane potential, and improves mitochondrial function. These findings suggest that kisspeptin may serve as a potential therapeutic target for treating insulin resistance in PCOS.

## Supporting information

**S1 Data. The minimal data set for this study has been uploaded as a supporting information file.**
(ZIP)

## Author contributions

**Writing – original draft:** Na Sun, Pingping Sun, Lilan Sun, Ping Sun, Dandan Qi, Huagang Ma.

**Writing – review & editing:** Na Sun, Huagang Ma.

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
