## [Decision Letter · Decision Letter 0]

20 Jul 2025

Dear Dr. Ma,

Thank you for submitting your manuscript to PLOS ONE. After careful consideration, we feel that it has merit but does not fully meet PLOS ONE’s publication criteria as it currently stands. Therefore, we invite you to submit a revised version of the manuscript that addresses the points raised during the review process.

**Please address all comments listed by two reviewers.**

We look forward to receiving your revised manuscript.

Kind regards,

Sheng Wu, Ph.D.

Academic Editor

PLOS ONE

Journal Requirements:

4. To comply with PLOS ONE submissions requirements, in your Methods section, please provide additional information regarding the experiments involving animals and ensure you have included details on methods of anesthesia and/or analgesia.

6. We note that your Data Availability Statement is currently as follows: If the data are all contained within the manuscript and/or Supporting Information files, enter the following: All relevant data are within the manuscript and its Supporting Information files.

7. Please update your submission to use the PLOS LaTeX template. The template and more information on our requirements for LaTeX submissions can be found at http://journals.plos.org/plosone/s/latex.

8. Thank you for stating the following financial disclosure:

Weifang Health Commission (Grand number: WFWSJK-2023-005)

9. We notice that your supplementary [tables] are included in the manuscript file. Please remove them and upload them with the file type 'Supporting Information'. Please ensure that each Supporting Information file has a legend listed in the manuscript after the references list.

10. Please include a copy of Table 2 which you refer to in your text on page 7.

Reviewers' comments:

Reviewer's Responses to Questions

**Comments to the Author**

1. Is the manuscript technically sound, and do the data support the conclusions?

Reviewer #1: Partly

Reviewer #2: Partly

2. Has the statistical analysis been performed appropriately and rigorously?

Reviewer #1: Yes

Reviewer #2: Yes

3. Have the authors made all data underlying the findings in their manuscript fully available?

Reviewer #1: No

Reviewer #2: Yes

4. Is the manuscript presented in an intelligible fashion and written in standard English?

Reviewer #1: Yes

Reviewer #2: Yes

Reviewer #1: The paper investigates the interaction between kisspeptin and insulin sensitivity in the ovaries, focusing on a PCOS mouse model. The study presents the concept and demonstrates that kisspeptin expression is altered in the ovaries of PCOS mice. Furthermore, external supplementation with kisspeptin enhanced GLUT4 expression, leading to improved glucose metabolism within the ovary. However the paper can be improved by doing the following:

1. Introduction:

Please elaborate on how insulin sensitivity is maintained or enhanced in the ovaries despite systemic insulin resistance in PCOS. Discuss the key insulin signaling pathways involved (e.g., PI3K-Akt, MAPK) and their differential regulation in ovarian versus peripheral tissues.

Additionally, the study reports downregulation of the Insulin- PI3K-Akt pathway in granulosa cells. However, some literature (e.g., PMID: 27060006) suggests insulin sensitivity in ovary under hyperandrogenic conditions with activation of insulin pathways. Please include this in the introduction.

2. Replace the term “kisspeptin antagonist” with “KISS1 receptor antagonist” or “KISS1R antagonist” throughout the manuscript for scientific accuracy.

3. In the Methods section, provide complete details of the in vivo model:

o Duration of DHEA treatment for follicle collection. References supporting the DHEA model. What was the rationale for using the DHEA model vs DHT?

o Time points for testosterone measurement following DHEA treatment.

o Was insulin supplemented to the culture media for the primary granulosa cells. Were experiments done with and without above.

o Please specify in methods and legends if kisspeptin and kisspeptin antagonist was added to PCOS-IR cells ?

o Catalog numbers for all reagents and kits used (e.g., testosterone and insulin kits, PMSG, goat serum)

4. In Figure 1A:

o Label the antral follicles and cysts clearly.

o Add data on the body weight of mice at or near the time of sacrifice.

5. In Figure 2, clarify whether the study rationale is based on local kisspeptin production in the ovary or on circulating levels. Since kisspeptin was added to the culture media, this likely simulates serum levels-please clarify in discussion. If Kisspeptin Receptor expression was measured by any means, please include it in the paper.

6. For Figure 3, specify the number of biological and technical replicates used. Include this information in the Methods section and figure legend. Also, though mentioned in the legend, it may be helpful to call the control group PCOS-IR, since Control is vehicle+ HFD group in figure 2. Add Data from Control-IR animals in supplementary if available, to see the independent effect of kisspeptin.

7. (Same as point 7) Ensure that replicate numbers are clearly indicated for all experiments in Figure 4.

8. In the Discussion, please comment on the observation that the rise in testosterone in the model group compared to controls is relatively modest, yet the difference in glucose tolerance test (GTT) results and HOMA-IR is quite significant. This may suggest aggravated insulin resistance in the presence of slightly increased testosterone. Also would be interesting to add some details on if insulin resistance preceeds morphological changes in the ovary.

Reviewer #2: The authors conclude that there is decreased kisspeptin production and that treating with kisspeptin improves insulin sensitivity. Did the authors measure serum kisspeptin levels? Ideally, serum kisspeptin levels would be reported at baseline and after HCG treatment for ovarian induction. Levels were measured after 24h of primary culture, so it is possible that expression could have changed during culture or in response to HCG stimulation.

The conclusions do to make it clear that all granulosa cell studies evaluating insulin resistance are in vitro studies. This needs to be clarified in the discussion and conclusions, particularly in the abstract. "Kisspeptin improves insulin resistance" in vitro in granulosa cells...

Page 18: misspelled "Previous"

**Do you want your identity to be public for this peer review?** For information about this choice, including consent withdrawal, please see our Privacy Policy

Reviewer #1: No

Reviewer #2: No

---

## [Author Response · Author response to Decision Letter 1]

23 Sep 2025

Answer: Thank you for your valuable suggestion. We will strictly follow the file naming convention you pointed out to name the files. Please review the files we have submitted.

Answer: We sincerely thank you for your valuable suggestion. As per your guidance, we have provided the original images for all Western blots. Please refer to the newly submitted files.

Answer: Thank you for your reminder. In accordance with your guidance, we have registered and verified our ORCID iD.

4. To comply with PLOS ONE submissions requirements, in your Methods section, please provide additional information regarding the experiments involving animals and ensure you have included details on methods of anesthesia and/or analgesia.

Answer: Thank you very much for your valuable suggestion. We have incorporated your instructions and added the method of euthanasia for mice in the Methods section of the manuscript. Please review the latest version we have provided.

5.Your ethics statement should only appear in the Methods section of your manuscript. If your ethics statement is written in any section besides the Methods, please delete it from any other section.

Answer: Thank you for your reminder. We have carefully checked the manuscript and ensured that the ethical statement is only presented in the Methods section. Any repeated ethical statements outside the Methods section have been removed accordingly.

6.We note that your Data Availability Statement is currently as follows: If the data are all contained within the manuscript and/or Supporting Information files, enter the following: All relevant data are within the manuscript and its Supporting Information files.

Answer: Thank you for your reminder. We have made the revisions as per your instructions.

Answer: Thank you for your reminder. Please refer to the latest supplementary information file we have provided.

7. Please update your submission to use the PLOS LaTeX template. The template and more information on our requirements for LaTeX submissions can be found at http://journals.plos.org/plosone/s/latex.

Answer: Thank you for your suggestion. We have submitted all the files as per your instructions.

8. Thank you for stating the following financial disclosure:

Weifang Health Commission (Grand number: WFWSJK-2023-005)

Answer: We sincerely thank you once again for your valuable suggestion. In accordance with your guidance, we have incorporated the supplementary explanation into the cover letter. Kindly review our latest revised manuscript.

9.We notice that your supplementary [tables] are included in the manuscript file. Please remove them and upload them with the file type 'Supporting Information'. Please ensure that each Supporting Information file has a legend listed in the manuscript after the references list.

Answer: We sincerely thank you once again for your valuable suggestion. In accordance with your guidance, the relevant content has been removed from the manuscript.

10. Please include a copy of Table 2 which you refer to in your text on page 7.

Answer: We sincerely thank you once again for your valuable suggestion. In accordance with your guidance, we have carefully reviewed the manuscript and corrected the content as Supplementary Table 2.

Answer: We sincerely thank you once again for your suggestion. We will review the references in accordance with your guidance.

Reviewers' comments:

Reviewer's Responses to Questions

Comments to the Author

1. Is the manuscript technically sound, and do the data support the conclusions?

Reviewer #1: Partly

Reviewer #2: Partly

2. Has the statistical analysis been performed appropriately and rigorously?

Reviewer #1: Yes

Reviewer #2: Yes

3. Have the authors made all data underlying the findings in their manuscript fully available?

Reviewer #1: No

Reviewer #2: Yes

4. Is the manuscript presented in an intelligible fashion and written in standard English?

Reviewer #1: Yes

Reviewer #2: Yes

5. Review Comments to the Author

Reviewer #1: The paper investigates the interaction between kisspeptin and insulin sensitivity in the ovaries, focusing on a PCOS mouse model. The study presents the concept and demonstrates that kisspeptin expression is altered in the ovaries of PCOS mice. Furthermore, external supplementation with kisspeptin enhanced GLUT4 expression, leading to improved glucose metabolism within the ovary. However the paper can be improved by doing the following:

1. Introduction:

Please elaborate on how insulin sensitivity is maintained or enhanced in the ovaries despite systemic insulin resistance in PCOS. Discuss the key insulin signaling pathways involved (e.g., PI3K-Akt, MAPK) and their differential regulation in ovarian versus peripheral tissues.

Answer: We sincerely thank you for your valuable suggestion. In accordance with your guidance, we have added the relevant information to the Introduction section of the manuscript. We kindly invite you to review the latest version of the manuscript.

Additionally, the study reports downregulation of the Insulin- PI3K-Akt pathway in granulosa cells. However, some literature (e.g., PMID: 27060006) suggests insulin sensitivity in ovary under hyperandrogenic conditions with activation of insulin pathways. Please include this in the introduction.

Answer: Thank you for your reminder. In accordance with your guidance, we have added the relevant references to the Introduction section of the manuscript. We kindly invite you to review the latest version of the manuscript.

2. Replace the term “kisspeptin antagonist” with “KISS1 receptor antagonist” or “KISS1R antagonist” throughout the manuscript for scientific accuracy.

Answer: We sincerely thank you for your valuable suggestion. In accordance with your guidance, we have made the necessary corrections in the manuscript. We kindly invite you to review the latest version.

3. In the Methods section, provide complete details of the in vivo model:

o Duration of DHEA treatment for follicle collection. References supporting the DHEA model. What was the rationale for using the DHEA model vs DHT?

Answer: We sincerely thank you for your valuable suggestion. In accordance with your guidance, we have added the relevant references regarding model establishment in the manuscript. We kindly invite you to review the latest version.

o Time points for testosterone measurement following DHEA treatment.

Answer: We sincerely thank you for your valuable suggestion. In accordance with your guidance, we have added clarification regarding the measurement time points in the manuscript.

o Was insulin supplemented to the culture media for the primary granulosa cells. Were experiments done with and without above.

Answer: We sincerely thank you for your valuable suggestion. In accordance with your guidance, we have corrected and clarified the culture medium for granulosa cells in the manuscript. We kindly invite you to review the latest version.

o Please specify in methods and legends if kisspeptin and kisspeptin antagonist was added to PCOS-IR cells ?

Answer: We sincerely thank you for your valuable suggestion. In accordance with your guidance, we have added the supplementary explanation in the manuscript. We kindly invite you to review the latest version.

o Catalog numbers for all reagents and kits used (e.g., testosterone and insulin kits, PMSG, goat serum)

Answer: We sincerely thank you once again for your valuable suggestion. In accordance with your guidance, we have added the supplementary information in the manuscript. We kindly invite you to review the latest version.

4. In Figure 1A:

o Label the antral follicles and cysts clearly.

Answer: We sincerely thank you once again for your suggestion. In accordance with your guidance, we have added the markings in Figure 1. We kindly invite you to review the latest version of the manuscript.

o Add data on the body weight of mice at or near the time of sacrifice.

Answer: We sincerely thank you once again for your valuable suggestion. We have added the time of mouse euthanasia in the manuscript. We kindly invite you to review the latest version.

5. In Figure 2, clarify whether the study rationale is based on local kisspeptin production in the ovary or on circulating levels. Since kisspeptin was added to the culture media, this likely simulates serum levels-please clarify in discussion. If Kisspeptin Receptor expression was measured by any means, please include it in the paper.

Answer: We thank the reviewer for this insightful comment. In response, we have clarified in the Discussion section that the rationale for our study is based on local ovarian production of kisspeptin, rather than circulating serum levels. Although kisspeptin was added to the culture media in vitro to mimic its physiological effects, the in vivo relevance is primarily due to its local synthesis within the ovary. Furthermore, we have now included information regarding kisspeptin receptor (KISS1R) expression in granulosa cells, which was measured and confirmed, in the revised manuscript. We kindly invite the reviewer to refer to the updated Discussion and Methods sections for these clarifications.

6. For Figure 3, specify the number of biological and technical replicates used. Include this information in the Methods section and figure legend. Also, though mentioned in the legend, it may be helpful to call the control group PCOS-IR, since Control is vehicle+ HFD group in figure 2. Add Data from Control-IR animals in supplementary if available, to see the independent effect of kisspeptin.

Answer: We sincerely thank you once again for your valuable suggestion. In accordance with your guidance, we have renamed the groups in the manuscript. We kindly invite you to review the latest version.

7. (Same as point 7) Ensure that replicate numbers are clearly indicated for all experiments in Figure 4.

Answer: We sincerely thank you once again for your valuable suggestion. We have clarified the number of experimental replicates in the Methods section and figure legends, and have corrected the group names in the manuscript. We kindly invite you to review the latest version.

8. In the Discussion, please comment on the observation that the rise in testosterone in the model group compared to controls is relatively modest, yet the difference in glucose tolerance test (GTT) results and HOMA-IR is quite significant. This may suggest a

---

## [Decision Letter · Decision Letter 1]

8 Oct 2025

Dear Dr. Ma,

Thank you for submitting your manuscript to PLOS ONE. After careful consideration, we feel that it has merit but does not fully meet PLOS ONE’s publication criteria as it currently stands. Therefore, we invite you to submit a revised version of the manuscript that addresses the points raised during the review process.

We look forward to receiving your revised manuscript.

Kind regards,

Sheng Wu, Ph.D.

Academic Editor

PLOS ONE

Journal Requirements:

Reviewers' comments:

Reviewer's Responses to Questions

**Comments to the Author**

Reviewer #1: (No Response)

2. Is the manuscript technically sound, and do the data support the conclusions?

Reviewer #1: Yes

3. Has the statistical analysis been performed appropriately and rigorously?

Reviewer #1: Yes

4. Have the authors made all data underlying the findings in their manuscript fully available?

Reviewer #1: Yes

5. Is the manuscript presented in an intelligible fashion and written in standard English?

Reviewer #1: Yes

Reviewer #1: (No Response)

**Do you want your identity to be public for this peer review?** For information about this choice, including consent withdrawal, please see our Privacy Policy

Reviewer #1: No

---

## [Author Response · Author response to Decision Letter 2]

27 Nov 2025

1.Fig 1A- There are no arrow marks. Please add them.

Answer: Thank you very much for your valuable suggestion. We sincerely apologize for the confusion caused by our oversight. In accordance with your instructions, we have replaced Figure 1, and we kindly invite you to review the updated version now provided in the revised manuscript.

2.Please indicate changes using coloring in the text.

Answer: Thank you for your feedback. We have highlighted all revised sections in the manuscript as instructed. Please review the latest version we have provided.

3.Reviewer question: Please indicate the line numbers for the addition regarding kisspeptin receptor expression in granulosa cells. I can’t find them.

“Answer: We thank the reviewer for this insightful comment. In response, we have clarified in the Discussion section that the rationale for our study is based on local ovarian production of kisspeptin, rather than circulating serum levels. Although kisspeptin was added to the culture media in vitro to mimic its physiological effects, the in vivo relevance is primarily due to its local synthesis within the ovary. “Furthermore, we have now included information regarding kisspeptin receptor (KISS1R) expression in granulosa cells, which was measured and confirmed, in the revised manuscript”.

Answer: We sincerely thank the reviewer for carefully checking our previous revision. We apologize for the oversight—although we stated in the last response that information regarding KISS1R expression in granulosa cells had been added, the corresponding text was not clearly incorporated in the revised manuscript, which caused confusion. We appreciate the reviewer’s patience in pointing this out.

In the current revision, we have corrected this issue and have now explicitly added the relevant description based on published evidence. The revised manuscript now includes the following statement in the Discussion section:

“In addition, previous studies have shown that kisspeptin is specifically expressed in granulosa cells, whereas its receptor KISS1R exhibits specific expression in oocytes.”

This information clarifies the cell-specific distribution of kisspeptin and its receptor in the ovary, thereby strengthening the mechanistic rationale of our study.

We appreciate the reviewer’s valuable comments, which have helped us improve the accuracy and clarity of the manuscript.

---

## [Editor Report · Decision Letter 2]

23 Dec 2025

Dear Dr. Ma,

Thank you for submitting your manuscript to PLOS ONE. After careful consideration, we feel that it has merit but does not fully meet PLOS ONE’s publication criteria as it currently stands. Therefore, we invite you to submit a revised version of the manuscript that addresses the points raised during the review process.

Request from the Editorial Office: We noted that the following related study was recently published by some of your coauthors:

Kisspeptin regulates the proliferation and apoptosis of ovary granulosa cells in polycystic ovary syndrome by modulating the PI3K/AKT/ERK signalling pathway (https://doi.org/10.1186/s12905-022-02154-6)

In order to comply with our Publication Criteria (http://journals.plos.org/plosone/s/criteria-for-publication#loc-2), please discuss this study in your Introduction and provide a sound scientific rationale for the submitted work. In your Discussion section, please discuss your findings in light of this existing paper. Please contact plosone@plos.org with any questions about this request. Thank you.

We look forward to receiving your revised manuscript.

Kind regards,

Sarah Jose, Ph.D.

Staff Editor

PLOS One
---

## [Author Response · Author response to Decision Letter 3]

30 Dec 2025

Journal Requirements:

1.If the reviewer comments include a recommendation to cite specific previously published works, please review and evaluate these publications to determine whether they are relevant and should be cited. There is no requirement to cite these works unless the editor has indicated otherwise.

Response: Thank you very much for your reminder.

Response: Thank you very much for your reminder.

---

## [Editor Report · Decision Letter 3]

7 Jan 2026

Dear Dr. Ma,

Thank you for submitting your manuscript to PLOS ONE. After careful consideration, we feel that it has merit but does not fully meet PLOS ONE’s publication criteria as it currently stands. Therefore, we invite you to submit a revised version of the manuscript that addresses the points raised during the review process.

**ACADEMIC EDITOR:**

We look forward to receiving your revised manuscript.

Kind regards,

Sheng Wu, Ph.D.

Academic Editor

PLOS One

---

## [Author Response · Author response to Decision Letter 4]

14 Jan 2026

Please specify the major differences between your article and the newly published study [Kisspeptin regulates the proliferation and apoptosis of ovary granulosa cells in polycystic ovary syndrome by modulating the PI3K/AKT/ERK signalling pathway (https://doi.org/10.1186/s12905-022-02154-6)]. The discussion is currently very general and superficial.

Answer: We sincerely thank you for your valuable suggestions. In accordance with your comments, we have comprehensively and systematically reorganized the Discussion section, and further clarified the similarities and differences between our study and the newly published literature. Please refer to the revised manuscript we have submitted.

---

## [Editor Report · Decision Letter 4]

19 Jan 2026

Kisspeptin improves local ovarian insulin resistance in PCOS by modulating the PI3K/AKT/GLUT4 signaling pathway

PONE-D-25-32818R4

Dear Dr. Ma,

We’re pleased to inform you that your manuscript has been judged scientifically suitable for publication and will be formally accepted for publication once it meets all outstanding technical requirements.

Kind regards,

Sheng Wu, Ph.D.

Academic Editor

PLOS One
---

## [Editor Report · Acceptance letter]

PONE-D-25-32818R4

PLOS One

Dear Dr. Ma,

I'm pleased to inform you that your manuscript has been deemed suitable for publication in PLOS One. Congratulations! Your manuscript is now being handed over to our production team.

Kind regards,

on behalf of

Dr. Sheng Wu

Academic Editor

PLOS One